# Bioshell Calcium Oxide-Containing Liquids as a Sanitizer for the Reduction of Histamine Production in Raw Japanese Pilchard, Japanese Horse Mackerel, and Chub Mackerel

**DOI:** 10.3390/foods9070964

**Published:** 2020-07-21

**Authors:** Sumiyo Hiruma, Masayuki Ishihara, Shingo Nakamura, Yoko Sato, Haruka Asahina, Koichi Fukuda, Tomohiro Takayama, Kaoru Murakami, Hidetaka Yokoe

**Affiliations:** 1Division of Biomedical Engineering, Research Institute, National Defense Medical College, 3-2 Namiki, Tokorozawa, Saitama 359-8513, Japan; iroihsh@gmail.com (S.H.); snaka@ndmc.ac.jp (S.N.); khf05707@nifty.com (K.F.); 2Division of Statistical Analysis, Research Support Center, Shizuoka General Hospital, 4-27-1 Kita-ando, Aoi-ku, Shizuoka 420-8527, Japan; sato.yoko.shiz@gmail.com; 3Department of Emergency Medicine, National Defense Medical College, 3-2 Namiki, Tokorozawa, Saitama 359-1234, Japan; h_asahina@ndmc.ac.jp; 4Department of Oral and Maxillofacial Surgery, National Defense Medical College, 3-2 Namiki, Tokorozawa, Saitama 359-8513, Japan; taka01@ndmc.ac.jp (T.T.); murakami@ndmc.ac.jp (K.M.); yokoe@ndmc.ac.jp (H.Y.)

**Keywords:** histamine, food poisoning, bioshell calcium oxides, bactericidal activity, disinfectant

## Abstract

Recently, there has been an increase in the number of food poisoning cases associated with histamine in food, mainly in relation to histamine in fish. Here, we investigated methods to decrease histamine levels in Japanese pilchard, Japanese horse mackerel, and chub Mackerel, stored at 10 °C using various concentrations of heated scallop bioshell calcium oxide (BiSCaO) suspension, dispersion (BiSCaO + Na_2_HPO_4_), colloidal dispersion (BiSCaO + NapolyPO_4_), scallop shell powder (SSP) Ca(OH)_2_ in pure water (PW) or saline, and BiSCaO water. BiSCaO in a high alkaline pH solution chemically decomposes histamine poorly, but the partial flocculation/precipitation of histamine was observed with 1 and 0.2 wt.% BiSCaO dispersion and BiSCaO colloidal dispersion, respectively. Cleaning fish samples with BiSCaO suspension, dispersion, colloidal dispersion, or BiSCaO water remarkably reduced histamine levels and normal bacterial flora (coliform bacteria (CF) and total viable bacterial cells (TC)) after storage for four days at 10 °C, while much higher histamine levels were observed after cleaning with saline. These results suggest that cleaning fish with BiSCaO dispersion, colloidal dispersion, or BiSCaO water can significantly reduce histamine levels through their bactericidal activity against histamine-producing bacteria.

## 1. Introduction

Histamine food poisoning is an allergy-like reaction caused by the consumption of fish or fermented foods containing a high concentration of histamine [1]. The clinical picture is usually characterized by a rapid onset (within 1 h) of symptoms, such as flushing, cutaneous rash, or headache; short duration; and self-limitation [2]. Histamine is produced by bacterial action during food processing and storage, and can be present in substantial amounts in blue-skinned fishes and fermented foodstuffs [3]. Therefore, the presence of histamine in foods is of great importance, as it acts as an indicator of the state of deterioration of the product, and is thus a potentially hazardous public health [4]. Indeed, histamine poisoning is one of the most common forms of toxicity caused by the ingestion of fish and fishery products [4,5,6,7]. Cooking, canning, or freezing do not reduce the levels of histamine, because this compound is heat stable. Humans are susceptible to histamine, and its effects can be described as intolerance or intoxication, depending on the severity of the symptoms [3]. The amount of histamine in food, the individual’s sensitivity, and their level of detoxification activity are the main factors affecting the toxicological response in consumers [3]. Classically, histamine toxicity is associated with eating blue-skinned fishes, such as Japanese pilchard (*Sardinops melanostictus*), Japanese horse mackerel (*Trachurus japonicus*), and chub mackerel (*Scomber japonicus*), that have been improperly stored after being caught. At warmer temperatures (>4 °C), the fish undergo overgrowth histamine-producing bacteria such as *Morganella morganii*, *Raoultella planticola*, and *Enterobacter aerogenes* [8,9,10]. *M. morganii* is a halophilic, low-temperature-growth bacterial species.

Blue-skinned fishes such as Japanese pilchard, Japanese horse mackerel, and chub mackerel contain a high level of the free amino acid histidine, a substrate for bacterial histidine decarboxylase that can form histamine from histidine. These bacteria can originate from normal fish flora (in particular, fish gut Enterobacterales), from the marine environment, and from the secondary contamination of food (improper handling, cross-contamination from catering equipment and facilities, and/or raw food) [5,6,7]. Histamine may also be produced during bacterial growth, and potentiate the histamine effect by inhibiting intestinal histamine-metabolizing enzymes [11]. The production of biogenic amines depends mainly on time-temperature abuse (deviation from the optimal storage temperature for a given time period), but also on fish product pH, salinity, oxygen availability, and competition with other spoilage microorganisms [12]. Histamine production is most commonly due to inadequate refrigeration of the fish, and can occur at any stage of the food chain [13].

Calcium oxide produced from limestone is an important inorganic compound used in various industries as, for example, an adsorbent, toxic-waste remediation agent, and alkalization agent. However, the calcium oxide from limestone is prepared by heating lime with oil or natural gas, and contains harmful impurities and has a dangerously high heat of hydration. In contrast, heated scallop shells using an electric furnace are an available source of CaO, and are used as an authorized food additive. However, most scallop shells are considered industrial waste, and the shells accumulate on the shores of harvesting districts in Japan, causing serious problems such as offensive odors and soil pollution. Therefore, we have chosen the safe CaO produced from scallop shells using an electric furnace (BiSCaO) as a part of resource recycling [14,15]. Heated scallop shell powder (SSP) is called bioshell calcium oxide (BiSCaO), and possesses deodorizing properties and a broad microbicidal activity against various pathogenic microbes, including viruses, bacteria, spores, and fungi. We previously reported that BiSCaO suspensions have a stronger bactericidal activity than hypochlorous acid (HOCl; pH 6) [16,17,18], sodium hypochlorite (NaClO; pH 8) [19,20], povidone iodine, and chlorhexidine gluconate (Hibiscrub®) solutions by using wood and pig skin pieces highly contaminated with normal bacterial flora (coliform bacteria (CF) and total viable bacterial cells (TC)) [21,22]. Furthermore, we developed a stable BiSCaO dispersion [14] and a colloidal dispersion [23] by adding phosphate compounds such as Na_2_HPO_4_ and NapolyPO_4_ to a BiSCaO suspension, respectively. The BiSCaO and colloidal dispersions showed higher deodorization and microbicidal activities than SSP-Ca(OH)_2_ and BiSCaO suspensions [11,20]. The objective of this study was to evaluate BiSCaO suspension, BiSCaO dispersion, BiSCaO colloidal dispersion, and BiSCaO water to reduce the histamine content of Japanese pilchard, Japanese horse mackerel, and chub mackerel.

## 2. Materials and Methods

### 2.1. BiSCaO Powder, Chemicals and Species of Blue-Skinned Fishes

Scallop shell powders heated at 1450 °C with an average BiSCaO particle diameter of 6 μm were purchased from Plus Lab Corp., Kanagawa, Japan. According to the manufacturer, the CaO concentration in all of the BiSCaO preparations exceeded 99%. SSP-Ca(OH)_2_ (Scallow, Kohkin Inst. Co. Ltd., Tochigi, Japan) had an average diameter of 3–6 μm, and CaO and Ca(OH)_2_ concentrations of <5% and >90%, respectively. Na_2_HPO_4_ (197-02865) and NapolyPO_4_ (PP) (694-05935) were purchased from FUJI FILM Wako Pure Chemical Corp., Osaka, Japan. Japanese pilchard (*Sardinops melanostictus*), Japanese horse mackerel (*Trachurus japonicus*), and chub mackerel (*Scomber japonicus*), as blue-skinned fishes, were purchased from a supermarket at Tokorozawa, Saitama, Japan.

### 2.2. BiSCaO Suspensions, BiSCaO Dispersion with Na_2_HPO_4_, BiSCaO Colloidal Dispersion with NapolyPO_4_ (PP) and BiSCaO Water

BiSCaO was added to saline (Otsuka pharmaceutical Co., Ltd. Tokyo, Japan) followed by rotary mixing to generate 1, 0.2, and 0.04 wt.% suspensions with saline, then 60% Na_2_HPO_4_ or 75% PP to BiSCaO was added to each BiSCaO suspension to produce 1, 0.2, and 0.04 wt.% BiSCaO dispersion or BiSCaO colloidal dispersion, respectively [14,23]. BiSCaO water was purchased from Plus Lab Corp., Kanagawa, Japan. According to the manufacturer, it was prepared by adding 10 wt.% BiSCaO to clean water and gently collecting the supernatant in a bottle. Another same volume of clean water was gently poured onto the BiSCaO precipitate, and the supernatant was gently collected in a bottle; this process was repeated 10 times. The BiSCaO water was colorless and transparent, with a pH of 12.8.

### 2.3. Removal of Histamine and Bactericidal Activity by BiSCaO Suspension, Dispersion, Colloidal Dispersion, and Water

Pure water (PW), produced with a water distillation apparatus (RFD250NB, Advantec. Co., Ltd., Tokyo, Japan), and contaminated saline (CS) were inoculated with normal bacterial flora (total viable cells (TC; about 8 × 10^6^ colony-forming units (CFUs) and coliform bacteria (CF; about 2 × 10^6^ CFUs)) by incubating the remaining warm water in a bathtub with 10% DMEM + 0.1 wt.% fetal bovine albumin (Wako Pure Chemical Corp.) + NaCl (0.9 wt.% ) at 37 °C for 24 h.

Histamine (084-00643, Wako Pure Chemical Corp.) was added to the PW and CS, and the concentrations of histamine were calculated according to instruction of a histamine test kit purchased from Kikkoman Biochemifa Company, Tokyo, Japan, which were 105.5 ± 11.5 ppm and 97.4 ± 10.1 ppm, respectively. The same volumes of 1, 0.2, and 0.04 wt.% suspension, dispersion, and colloidal dispersion of BiSCaO and SSP-Ca(OH)_2_, and BiSCaO water were added to the PW and CS containing added histamine, followed by rotary mixing for 10 min. After centrifugation (1000 rpm, 190× *g*) for 5 min, the levels of histamine in the supernatants were measured using a histamine test kit (Kikkoman Biochemifa Company, Kanagawa, Japan). The range of detection was 0.4–6.0 ppm (10–150 ppm when fish meat is diluted 25-fold). Although it has not been described in the text, detailed information for the kit is available at https://biochemifa.kikkoman.co.jp/e/products/detail/?id=11160.

Then, 10 mL of 2, 0.4, and 0.08 wt.% suspension, dispersion, colloidal dispersion of BiSCaO and SSP-Ca(OH)_2_, or BiSCaO water, were added to 10 mL of CS; mixed well; and then incubated at room temperature for 15 min. The number of CFUs per sample was determined by gently pouring aliquots (1 mL of each supernatant) into individual Petri dishes containing pre-aliquoted portions of dry medium for total viable bacterial cells (TC) and coliform bacteria (CF) (Compact Dry, Nissui Pharmaceutical Co., Ltd., Tokyo, Japan) [24,25], and incubating the plates for 24 h in a 37 °C incubator (Alp Co., Ltd., Tokyo, Japan).

### 2.4. Reduction of Histamine Using Decomposing Raw Japanese Pilchard, Japanese Horse Mackerel and Chub Mackerel, and Reduction of Bactericidal Activity by BiSCaO Suspension, Dispersion, Colloidal Dispersion, and BiSCaO Water

Raw slices (about 50 g) of the blue-skinned fishes Japanese pilchard, Japanese horse mackerel, and chub mackerel were rinsed with 100 mL of BiSCaO (0.2 wt.%) suspension, dispersion, colloidal dispersion, BiSCaO water (undiluted) or SSP-Ca(OH)_2_ (0.2 wt.%) as disinfectants (*n* = 3). Comparable slices were similarly rinsed with saline as the controls. The slices rinsed with each disinfectant were incubated at 10 °C in a Cool Incubator (A1201; Ikuta Sangyo, Co. Ltd., Nagano, Japan) for 2, 4, and 6 days. Furthermore, the raw slices were rinsed with saline and pre-incubated at 10 °C for 3 days; rinsed with 100 mL of BiSCaO (0.2 wt.%) suspension, dispersion, colloidal dispersion, or BiSCaO water (undiluted) or SSP-Ca(OH)_2_ (0.2 wt.%) as disinfectants; then incubated at 10 °C for 1 h, 1 day, and 2 days.

The samples were boiled for 10 min at 100 °C, cooled to room temperature, centrifuged at 1000 rpm (190× *g*) for 5 min, and the histamine content of each was determined by measuring OD_470_ using a histamine test kit (Kikkoman Biochemifa Company), according to manufacturer’s instructions (*n* = 3).

At each time point, a piece of Japanese horse mackerel slice was gently rinsed with saline, minced, and strongly vortexed for 1 min with 50 mL saline. A small aliquot of each sample (1 mL) was centrifuged at 1000 rpm (190× *g*) for 5 min, and the TC and CF values of the supernatants were determined. The number of CFUs were determined in 1 mL aliquots of each supernatant, as described above.

## 3. Results

### 3.1. Removal of Histamine and Bactericidal Activity Using BiSCaO Suspension, Dispersion, Colloidal Dispersion, SSP-Ca(OH)_2_, or BiSCaO Water

We evaluated the efficiency of the removal of added histamine by 0.04, 0.2, and 1.0 wt.% BiSCaO suspension, dispersion, colloidal dispersion, SSP-Ca(OH)_2_, or undiluted BiSCaO water in pure water (PW) or contaminated saline (CS), followed by rotary mixing for 10 min and re-measuring the levels of histamine in the supernatants using a histamine test kit. The histamine concentrations of the control in the PW and CS were 105.5 ± 11.5 ppm and 97.4 ± 10.1 ppm, respectively.

The tested samples in PW and CS at high alkaline pH poorly decomposed histamine, but histamine can be partially flocculated/precipitated with 1 and 0.2 wt.% BiSCaO dispersion and by BiSCaO colloidal dispersion. BiSCaO water (undiluted) and 1 wt.% BiSCaO suspension in CS can also weakly flocculate/precipitate histamine (Table 1). Thus, although BiSCaO-based disinfectants directly decompose histamine little, BiSCaO dispersion and colloidal dispersion might partially remove histamine from CS by flocculation/precipitation [22].

### 3.2. Microbicidal Efficacy of BiSCaO Suspension, Dispersions, Colloidal Dispersion, and BiSCaO Water

An equal volume of CS was mixed with either BiSCaO suspension, dispersion, colloidal dispersion, SSP-Ca(OH)_2_, or BiSCaO water in CS; mixed well; incubated at room temperature for 15 min; then, the number of TC and CF (CFUs) per sample was determined. The 1 wt.% BiSCaO suspension, dispersion, colloidal dispersion, SSP-Ca(OH)_2_ suspension, and BiSCaO water (undiluted and 2-fold diluted) decreased the total viable bacterial cells (TC) and coliform bacteria (CF) to below the limit of detection (Table 2). SSP-Ca(OH)_2_ suspension, dispersion, and colloidal dispersion at 0.2 and 0.04 wt.% had a lower microbicidal activity than the corresponding BiSCaO preparations.

### 3.3. Reduction of Histamine in Decomposing Raw Japanese Pilchard, Japanese Horse Mackerel and Chub Mackerel, and Bactericidal Activity of BiSCaO Suspension, Dispersion, Colloidal Dispersion and BiSCaO Water

Raw slices of Japanese pilchard, Japanese horse mackerel, and chub mackerel were cleaned with BiSCaO suspension, dispersion, or colloidal dispersion in saline, BiSCaO water (undiluted), or SSP-Ca(OH)_2_ suspension as disinfectants, then these slices with residual disinfectants were incubated at 10 °C for the indicated periods (Table 3). Histamine is the only biogenic amine with regulatory limits set by European Legislation, up to a maximum of 200 ppm in fresh fish, and it rises in improperly stored raw fish to over 200 ppm, which may be associated with clinical illness [5,7,23]. Therefore, it is inferred that the histamine threshold for safe raw fish consumption is 200 ppm.

The histamine levels of samples were below 50 ppm on day 1, and were below 100 ppm on day 2, except for the saline-cleansed sample (control) in Japanese pilchard, whereas the histamine of Japanese pilchard, Japanese horse mackerel, and chub mackerel in control on day 4 were 880 ± 180, 460 ± 68, and 235 ± 36 ppm, respectively, which exceeded the histamine threshold for safe raw fish consumption. Furthermore, the histamine of Japanese pilchard, Japanese horse mackerel, and chub mackerel in the SSP-Ca(OH)_2_ suspension on day 4 were 232 ± 56 and 207 ± 35 ppm, respectively. In contrast, the BiSCaO dispersion, colloidal dispersion, and BiSCaO water remarkably reduced histamine levels to below 60 ppm in all three sample types, especially on day 4, compared to the control and SSP-Ca(OH)_2_ suspension (Table 3).

The histamine levels of pre-incubated Japanese pilchard, Japanese horse mackerel, and chub mackerel in saline (control) at 10 °C for 3 days were 240 ± 42 ppm (Japanese pilchard), 184 ± 32 ppm (Japanese horse mackerel), and 158 ± 25 ppm (chub mackerel). Histamine levels were significantly reduced 1 h after cleaning these samples with saline (control), BiSCaO suspension, dispersion, colloidal dispersion, BiSCaO water, or SSP-Ca(OH)_2_ suspension (Table 4). Cleaning with BiSCaO dispersion, colloidal dispersion, or BiSCaO water more effectively reduced histamine levels than saline, BiSCaO suspension, or SSP-Ca(OH)_2_ suspension. After cleaning with saline, BiSCaO suspension, or SSP-Ca(OH)_2_ suspension, the histamine levels were higher after 1 and 2 days compared with the BiSCaO dispersion, colloidal dispersion, or BiSCaO water (Table 4). In particular, the histamine levels 2 days after cleaning with BiSCaO dispersion, colloidal dispersion, or BiSCaO water were 100–200 ppm, which were below the histamine threshold for safe raw fish consumption.

### 3.4. Antimicrobial Efficacy of BiSCaO Suspension, Dispersions, Colloidal Dispersion and BiSCaO Water Determined Using pre-Incubated Japanese Horse Mackerel Slices

Raw Japanese horse mackerel samples cleaned with disinfectants of BiSCaO suspension, dispersion, colloidal dispersion, SSP-Ca(OH)_2_, and BiSCaO water were incubated at 10 °C for preset periods. At each time point, a small aliquot (1 g) of Japanese horse mackerel sample (about 50 g) incubated at 10 °C was added to 10 mL of either 0.2 wt.% BiSCaO suspension, dispersion, colloidal dispersion, SSP-Ca(OH)_2_, or BiSCaO water; vortexed well; incubated at room temperature for 15 min; and then the number of TC and CF (CFUs) per sample were determined. The TC and CF counts in the saline (control) and SSP-Ca(OH)_2_ group were greatly increased compared with the other samples, whereas BiSCaO dispersion, colloidal dispersion, and BiSCaO water markedly reduced the TC and CF counts compared with BiSCaO suspension and SSP-Ca(OH)_2_ suspension (Figure 1).

The TC and CF levels of Japanese horse mackerel samples pre-incubated with saline at 10 °C for 3 days were both 82 ± 5 (×10^3^). CFUs TC and CF were significantly reduced 1 h, 1 day, and 2 days after cleaning with BiSCaO suspension, dispersion, colloidal dispersion, BiSCaO water, or SSP-Ca(OH)_2_ suspension, and the BiSCaO dispersion, colloidal dispersion, and BiSCaO water were more effective than the saline, BiSCaO suspension, and SSP-Ca(OH)_2_ suspension (Figure 2). Thus, the cleaning of Japanese horse mackerel with BiSCaO dispersion, colloidal dispersion, or BiSCaO water was more effective at decreasing TC and CF than saline, BiSCaO suspension, and SSP-Ca(OH)_2_ suspension.

## 4. Discussion

Histamine is typically less than 1 ppm in properly stored fish, but rises in contaminated fish to 200–500 ppm, which is associated with clinical illness, and is the only biogenic amine with regulatory limits set by European Legislation, namely, up to a maximum of 200 ppm in fresh fish and 400 ppm in fishery products treated by enzyme maturation in brine [5,7,26]. A safe and efficient cleaning method for blue-skinned fishes, such as Japanese pilchard, Japanese horse mackerel, and chub mackerel, is required to reduce thehistamine level and histamine-producing bacteria.

BiSCaO dispersion, colloidal dispersion, and BiSCaO water in PW at a high alkaline pH poorly decompose histamine, but histamine can be partially flocculated/precipitated with 1 and 0.2 wt.% BiSCaO dispersion and by BiSCaO colloidal dispersion in a contaminated solution (CS). BiSCaO directly degrades histamine a little, but BiSCaO dispersion and colloidal dispersion might partially remove histamine from CS by flocculation/precipitation.

The histamine levels of Japanese pilchard, Japanese horse mackerel, and chub mackerel samples incubated at 10 °C for 4 days after cleaning with saline increased exponentially. In contrast, BiSCaO dispersion, colloidal dispersion, and BiSCaO water reduced histamine levels, especially on days 2 and 4. The histamine levels of Japanese pilchard, Japanese horse mackerel, and chub mackerel samples pre-incubated with saline at 10 °C for 3 days were above 150 ppm. The histamine levels were significantly reduced 1 h after cleaning with saline (control), BiSCaO suspension, dispersion, colloidal dispersion, BiSCaO water, or SSP-Ca(OH)_2_ suspension. Cleaning with BiSCaO dispersion, colloidal dispersion, or BiSCaO water reduced the histamine more significantly than saline, BiSCaO suspension, and SSP-Ca(OH)_2_ suspension 2 days after cleaning. Thus, cleaning Japanese pilchard, Japanese horse mackerel, and chub mackerel fishes with BiSCaO-containing liquids such as BiSCaO dispersion, colloidal dispersion, or BiSCaO water is effective at reducing histamine levels for 4–5 days, even in the absence of adequate refrigeration.

The bacterial infection of food can be prevented by proper cleaning with disinfectants rather than with water or saline. However, the disinfectants used clinically, such as povidone-iodine and chlorhexidine gluconate, are cytotoxic and expensive [27], and high concentrations are required for bactericidal activity [28,29]. This has led to research on various disinfectants, such as silver nanoparticles [30,31,32], bioshell calcium oxide (BiSCaO) powder, hypochlorous acid (HClO), and sodium hypochlorite (NaClO), all of which exhibit bactericidal properties at relatively low concentrations [16,17,18].

Cleaning Japanese horse mackerel with BiSCaO-containing liquids is effective for reducing TC and CF and histamine levels for 4–5 days, even in the absence of adequate refrigeration. These results indicate that BiSCaO disprsion, colloidal dispersion, and BiSCaO water reduced histamine level in the fishes, and they may reduce histamine-producing bacteria such as *Morganella morganii*, without directly degrading histamine. However, as this study did not prove that the BiSCaO solution reduced histamine-producing bacteria such as *Morganella morganii*, it is required to identify the microbial species responsible for high levels of histamine production in blue-skinned fishes and to reduce histamine-producing bacteria in future.

## 5. Conclusions

Although BiSCaO-containing liquids at a high alkaline pH decomposes little histamine, cleaning blue-skinned fishes such as Japanese pilchard, Japanese horse mackerel, and chub mackerel with BiSCaO suspension, dispersion, colloidal dispersion, or BiSCaO water reduces histamine levels through bactericidal activity against histamine-producing bacteria.

## Figures and Tables

**Figure 1 foods-09-00964-f001:**
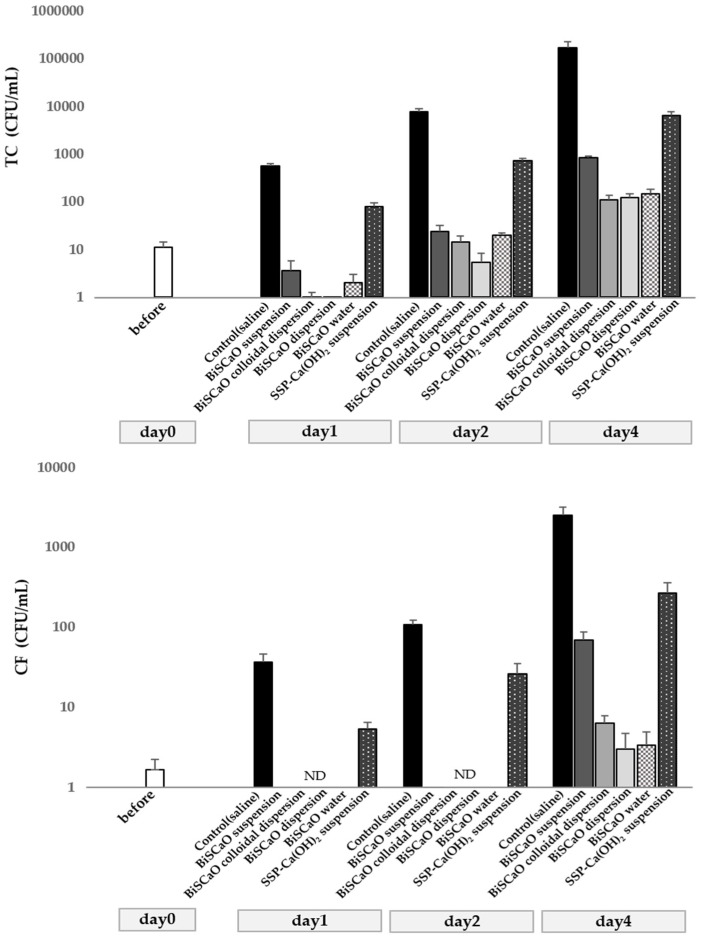
Anti-microbial efficacy of BiSCaO-based disinfectants. The number of CFU/mL of TC (**upper**) and CF (**lower**) in 1 g of raw Japanese horse mackerel cleaned with each BiSCaO disinfectant after incubation for 1, 2, and 4 days were counted after vigorous vortexing with 10 mL of pure water for 30 s (*n* = 3). Pure water was used as the control. ND indicates not detected.

**Figure 2 foods-09-00964-f002:**
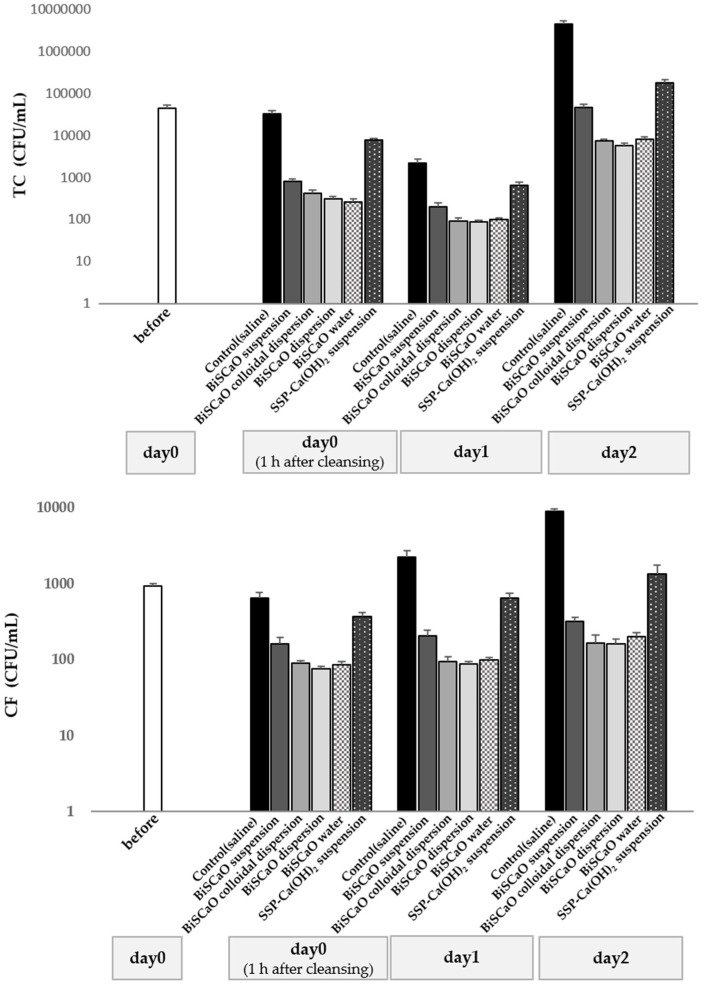
Anti-microbial efficacy of BiSCaO-based disinfectants. The number of CFU/mL of TC (**upper**) and CF (**lower**) in 1 g of raw Japanese horse mackerel pre-incubated with saline at 10 °C for 3 days was counted 1 h, 1 day, and 2 days after cleaning with each BiSCaO disinfectant and vigorously vortexing with 10 mL of pure water for 30 s (*n* = 3). Pure water was used as the control.

**Table 1 foods-09-00964-t001:** Reduction of histamine concentration (ppm) in pure water (PW) and contaminated saline (CS) by various BiSCaO-based disinfectants (*n* = 3).

	Reagent Concentration						
	1 wt.%		0.2 wt.%		0.04 wt.%		
Test Sample	PW	CS	PW	CS	PW	CS	
BiSCaO suspension	47.5 ± 3.5 12.6 ± 0.03	31.3 ± 0.9 12.5 ± 0.02	51.2 ± 3.7 12.3 ± 0.01	35.2 ± 2.7 12.2 ± 0.01	76.2 ± 8.5 11.5 ± 0.01	67.1 ± 7.0 11.2 ± 0.02	(ppm) pH
BiSCaO colloidal dispersion	21.9 ± 1.8 12.6 ± 0.02	18.1 ± 1.0 12.4 ± 0.02	29.2 ± 2.4 12.5 ± 0.02	24 ± 1.3 12.2 ± 0.01	67.1± 4.7 11.7 ± 0.02	62.5 ± 6.0 11.5 ± 0.01	(ppm) pH
BiSCaO dispersion	26.8 ± 2.6 12.6 ± 0.02	21.9 ± 1.4 12.3 ± 0.01	42.1 ± 3.9 12.3 ± 0.02	32.6 ± 3.4 12.0 ± 0.01	70.3 ± 6.9 11.5 ± 0.01	66.8 ± 7.6 11.1 ± 0.02	(ppm) pH
SSP-Ca(OH)_2_ suspension	71.5 ± 7.3 12.2 ± 0.01	68.2 ± 6.9 11.9 ± 0.01	85.1 ± 8.4 12.0 ± 0.01	73.7 ± 6.4 11.8 ± 0.02	91.5 ± 9.3 11.5 ± 0.01	88.8 ± 8.3 11.2 ± 0.01	(ppm) pH
SSP-Ca(OH)_2_ colloidal dispersion	59.5 ± 3.7 12.2 ± 0.01	35.5 ± 3.6 11.9 ± 0.01	74.1 ± 7.1 12.0 ± 0.01	61.1 ± 6.3 11.8 ± 0.01	88.7 ± 8.2 11.5 ± 0.01	76.3 ± 6.5 11.2 ± 0.01	(ppm) pH
SSP-Ca(OH)_2_ dispersion	61.9 ± 6.1 12.4 ± 0.01	41.3 ± 3.7 12.2 ± 0.01	75.2 ± 7.7 12.0 ± 0.01	64.4 ± 6.9 11.6 ± 0.01	90.2 ± 9.0 11.3 ± 0.01	81.6 ± 8.0 10.8 ± 0.01	(ppm) pH
	**Reagent Concentrations**						
	**Undiluted**		**2-fold diluted**		**4-fold diluted**		
**Test sample**	**PW**	**CS**	**PW**	**CS**	**PW**	**CS**	
BiSCaO water	50.9 ± 5.0 12.7 ± 0.03	27.7 ± 2.4 12.6 ± 0.02	55.1 ± 4.1 12.6 ± 0.02	35.3 ± 2.9 12.5 ± 0.02	72.6 ± 8.8 12.2 ± 0.03	68.5 ± 7.4 12.0 ± 0.01	(ppm) pH

Results are presented as means and standard deviations.

**Table 2 foods-09-00964-t002:** Decrease of total viable bacterial cells (TC) and coliform bacteria (CF) by various BiSCaO-based disinfectants (*n* = 3).

	Reagent Concentration					
	1 wt.%		0.2 wt.%		0.04 wt.%	
Test Sample	TC	CF	TC	CF	TC	CF
BiSCaO suspension	0	0	84 ± 12	15 ± 5	68 ± 17 (×10^3^)	25 ± 6 (×10^3^)
BiSCaO colloidal dispersion	0	0	16 ± 5	3 ± 2	72 ± 21 (×10^2^)	18 ± 5 (×10^2^)
BiSCaO dispersion	0	0	44 ± 10	6 ± 2	12 ± 4 (×10^3^)	42 ± 11 (×10^2^)
SSP-Ca(OH)_2_ suspension	0	0	9 ± 3 (×10^2^)	21 ± 5 (×10)	25 ± 6 (×10^5^)	48 ± 12 (×10^4^)
SSP-Ca(OH)_2_ colloidal dispersion	0	0	32 ± 7 (×10)	56 ± 16	68 ± 16 (×10^4^)	81 ± 32 (×10^3^)
SSP-Ca(OH)_2_ dispersion	0	0	48 ± 9 (×10)	62 ± 18	87 ± 23 (×10^4^)	12 ± 5 (×10^4^)
	**Reagent Concentration**					
	**Undiluted**		**2-fold diluted**		**4-fold diluted**	
**Test sample**	**CF**	**TC**	**CF**	**TC**	**CF**	**TC**
BiSCaO water	0	0	0	0	88 ± 21 (×10)	27 ± 7 (×10)

The number shows colony-forming unit (CFU)/mL of total viable bacterial cells (TC) and coliform bacteria (CF). Results are presented as means and standard deviations.

**Table 3 foods-09-00964-t003:** Reduction of histamine in Japanese pilchard, Japanese horse mackerel, and chub mackerel after cleaning with various BiSCaO-based disinfectants (*n* = 3).

**Day 1**			
	**Histamine** (ppm)		
**Test sample**	Japanese pilchard	Japanese horse mackerel	Chub mackerel
Control (saline)	40 ± 5	25 ± 3	19 ± 2
BiSCaO suspension	12 ± 2	9 ± 1	8 ± 1
BiSCaO colloidal dispersion	10 ± 2	7 ± 1	4 ± 1
BiSCaO dispersion	9 ± 1	6 ± 1	5 ± 1
BiSCaO water	11 ± 2	8 ± 2	5 ± 1
SSP-Ca(OH)_2_ suspension	19 ± 5	14 ± 3	10 ± 2
**Day 2**			
	**Histamine** (ppm)		
**Test sample**	Japanese pilchard	Japanese horse mackerel	Chub mackerel
Control (saline)	168 ± 21	95 ± 22	43 ± 10
BiSCaO suspension	31 ± 6	26 ± 6	18 ± 6
BiSCaO colloidal dispersion	15 ± 3	18 ± 5	11 ± 2
BiSCaO dispersion	14 ± 3	18 ± 5	11 ± 2
BiSCaO water	15 ± 3	20 ± 6	12 ± 2
SSP-Ca(OH)_2_ suspension	56 ± 12	65 ± 15	23 ± 6
**Day 4**			
	**Histamine** (ppm)		
**Test sample**	Japanese pilchard	Japanese horse mackerel	Chub mackerel
Control (saline)	880 ± 180	460 ± 68	235 ± 36
BiSCaO suspension	92 ± 19	70 ± 16	59 ± 9
BiSCaO colloidal dispersion	32 ± 5	55 ± 8	27 ± 6
BiSCaO dispersion	38 ± 7	57 ± 8	28 ± 8
BiSCaO water	41 ± 8	59 ± 9	27 ± 7
SSP-Ca(OH)_2_ suspension	232 ± 56	207 ± 35	90 ± 17

Results are presented as means and standard deviations.

**Table 4 foods-09-00964-t004:** Reduction of histamine in pre-incubated Japanese pilchard, Japanese horse mackerel, and chub mackerel samples by cleaning with various BiSCaO-based disinfectants (*n* = 3).

**1 h after Cleansing**			
	**Histamine** (ppm)		
**Test sample**	Japanese pilchard	Japanese horse mackerel	Chub mackerel
Control (saline)	195 ± 31	140 ± 22	108 ± 12
BiSCaO suspension	115 ± 22	97 ± 11	66 ± 11
BiSCaO colloidal dispersion	96 ± 17	86 ± 10	45 ± 9
BiSCaO dispersion	93 ± 15	84 ± 11	44 ± 10
BiSCaO water	102 ± 23	80 ± 9	50 ± 8
SSP-Ca(OH)_2_ suspension	136 ± 18	110 ± 19	57 ± 12
**1 day after Cleansing**			
	**Histamine** (ppm)		
**Test sample**	Japanese pilchard	Japanese horse mackerel	Chub mackerel
Control (saline)	451 ± 32	305 ± 18	212 ± 15
BiSCaO suspension	145 ± 25	121 ± 16	92 ± 15
BiSCaO colloidal dispersion	112 ± 16	95 ± 10	65 ± 9
BiSCaO dispersion	110 ± 10	91 ± 7	61 ± 8
BiSCaO water	125 ± 21	98 ± 8	66 ± 10
SSP-Ca(OH)_2_ suspension	265 ± 25	210 ± 15	142 ± 16
**2 day after Cleansing**			
	**Histamine** (ppm)		
**Test sample**	Japanese pilchard	Japanese horse mackerel	Chub mackerel
Control (saline)	920 ± 180	790 ± 87	458 ± 58
BiSCaO suspension	311 ± 56	210 ± 22	167 ± 9
BiSCaO colloidal dispersion	145 ± 27	111 ± 24	105 ± 12
BiSCaO dispersion	140 ± 28	105 ± 15	106 ± 11
BiSCaO water	161 ± 31	121 ± 12	110 ± 14
SSP-Ca(OH)_2_ suspension	468 ± 98	432 ± 78	335 ± 23

Results are presented as means and standard deviations.

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
