# Peer review of "Bioshell Calcium Oxide-Containing Liquids as a Sanitizer for the Reduction of Histamine Production in Raw Japanese Pilchard, Japanese Horse Mackerel, and Chub Mackerel"

_foods, 2020, doi:10.3390/foods9070964_

Round 1

Reviewer 1 Report

General remarks:

This article is well written and has interesting results that can make a good contribution for the study of the reduction of histamine production in fish.

In order to increase the manuscript clarity, there are some issues that need to be addressed

To facilitate the location of my comments, I am going to present them organized by line number of the “foods-841496-peer-review-v1” pdf file.

Revision comments:

Line 2: The title does not describe adequately the work done in this study. Prior to apply BiSCaO to the fish samples (method 2.4), there were 3 other methods (methods 2.1, 2.2. and 2.3) that tried to show the potential and properties of BiSCaO in vitro experiments (without fish). The title should also include these aspects of the work. The title should also include the names of the fish species used in the study. “Blue-skinned fish” is an enormous group of different species, and this work only focus on 3 taxa.

Line 21: There are several different species that are non-scientifically referred as “sardine” and “horse mackerel” depending on the geographical site of sampling. “Scombroid” is an ambiguous word, that defined in the past some species of the Scombridae Family. The name “scombroid” is also associated with “Scombroid food poisoning”, which is an old way of referring to histamine food poisoning. The are fish species that have high values of histamine, that do not belong to the Scombridae Family. This issue related to the identification of the species used in this work, must be clarified through all the manuscript!

Line 22-24: The abstract is a bit confusing. The authors state only using 0.2 wt% BiSCaO, but afterwards they refer using other concentration…

Lines 39-41: This sentence needs a citation.

Lines 51-54: Such an important statement should be reinforced with more than 1 citation.

Lines 70-72: The objective is too generic and does not reflect all the work presented afterwards.

Line 74: Why do authors choose to use CaO obtained from scallop shells, and not from limestone rocks? Is there any sustainability or circular economy issues associated with this work? If so, this issue should be explained in the introduction. If not, authors should explain why scallop CaO is better then other types of CaO.

Line 99: Authors must identify which histamine kit they used and its precision. If this work was done onboard a vessel, I could understand the use of histamine kits, but in inland laboratories why did not the authors used HPLC methods to determine histamine concentrations?

Line 113: How many replicates were done in each sampling moment? Indicate the scientific names of species used in the study.

Line 127: No statistical methods were used to compare treatments? Did the authors used triplicates in all sampling moments? Are the results presented as means and standard deviations?

Line 142: The table is confusing, not being clear what is histamine concentrations and pH values. The histamine results are means and standard deviations? Why is the pH just a single value? No replicates?

Line 153: Are the results means and standard deviations or standard error? Presenting the results this way, makes it difficult to compare absolute values and the effects of the treatments.

Line 169: Are these histamine values high or low? Which are the legal limits of histamine concentrations in fish? Which is the histamine threshold for safe fish consumption?

Line 172: Figure 1 is just a repetition of the horse mackerel data from Table 3. I suggest converting Table 3 into figures, like Figure 1, and refer the exact values of histamine concentrations when describing the results in the text. In Figure 1, bars are means, and error bars are standard deviations?

Line 190: Why only graph for horse mackerel?

Lines 196-222: The results are lacking all the data about sardines and scombroid? The Methods do not state that this part of the work was only done for horse mackerel…

Line 237: The word “remarkably” is not scientifically appropriate! Authors should present some statistical data about percentage of reduction for histamine formation in each treatment.

Lines 252-264: This part of the discussion is only a description of the results. There is no discussion of results!

Line 265: This work does not prove that BiSCaO water can “kill” bacteria, it only shows that bacteria did not grow to form colonies. This work does not prove that bacteria such as Morganella morganii were killed…authors should refrain from making assumptions not supported by their results!

Lines: 269-271: The first two sentences of the Conclusions are just general statements, not supported by this study results

Line 273: See comment about line 237

Reviewer 2 Report

Revision Paper: Application of bioshell calcium oxide (BiSCaO) for the reduction of histamine production in raw blue skinned fish samples

Revision Paper: Application of bioshell calcium oxide (BiSCaO) for the reduction of histamine production in raw blue skinned fish samples

Sumiyo Hiruma, Masayuki Ishihara, Shingo Nakamura, Yoko Sato, Haruka Asahina, Koichi

Fukuda, Tomohiro Takayama, Kaoru Murakami and Hidetaka Yokoe

In this manuscript, the authors describe the study of procedures used to decrease the levels of histamine in three species of blue-skinned fish like sardine, scombroid fish and horse mackerel. Concluding that BiSCaO dispersion, colloidal dispersion, or BiSCaO water can significantly reduce histamine levels through their bactericidal activity against histamine-producing bactéria in stored fish. Recognizing the value of the contribution given by the authors, I address the following comments/suggestions to improve the value of the paper.

As an English non-native speaker, I understand the difficulties of writing in a language distinct from our mother tongue. I would recommend a minor English revision of the manuscript since it showed some small errors. The majority of the references are updated which is a value point.

Comments/Improvement corrections:

Introduction Section

Page 2 line 44: The sentence needs to be supported with reference.

Materials and Methods Section

Page 3 line 93: ….”Apparatus”…

For clarity sake, used species should be referenced in the methods section.

LOQ and LOD of the test kit should be specified.

Discussion

Page 10 1st paragraph: Should be moved to the introduction section.

Authors should state the regulatory limits of histamine in fish products, as well as state the safety/ advantage of the use of BiSCaO and it’s suitability as a cleansing method. In that way, strengthening their work.

Round 2

Reviewer 1 Report

The authors did a very good work in this new version.

I only have one comment that need a small correction:

- In the bottom part of all Tables the sentence "Those result represent standard deviations", should be replaced by "Results are presented as means and standard deviations".

Author Response

Reviewer #1

Comments and Suggestions for Authors

The authors did a very good work in this new version.

I only have one comment that need a small correction:

- In the bottom part of all Tables the sentence "Those result represent standard deviations", should be replaced by "Results are presented as means and standard deviations".

Thank you very much for pointing out it. We corrected it (Table 1-4). Furthermore, we performed error editing throughout TEXT.
